# Structural and Functional Insights into Dishevelled-Mediated Wnt Signaling

**DOI:** 10.3390/cells13221870

**Published:** 2024-11-11

**Authors:** Lei Wang, Rui Zhu, Zehua Wen, Hua-Jun Shawn Fan, Teresa Norwood-Jackson, Danielle Jathan, Ho-Jin Lee

**Affiliations:** 1College of Chemical Engineering, Sichuan University of Science and Engineering, Zigong 643000, China; 322086001105@stu.suse.edu.cn (L.W.); zhurui0723@icloud.com (R.Z.); zehuawen5733@163.com (Z.W.); fan27713@yahoo.com (H.-J.S.F.); 2Division of Natural & Mathematical Sciences, LeMoyne-Owen College, Memphis, TN 38126, USA; tnorwood_jackson295@loc.edu (T.N.-J.); djathan240@loc.edu (D.J.)

**Keywords:** Dishevelled, Wnt, mechanism, AlphaFold, post-translation, cancer

## Abstract

Dishevelled (DVL) proteins precisely control Wnt signaling pathways with many effectors. While substantial research has advanced our understanding of DVL’s role in Wnt pathways, key questions regarding its regulatory mechanisms and interactions remain unresolved. Herein, we present the recent advances and perspectives on how DVL regulates signaling. The experimentally determined conserved domain structures of DVL in conjunction with AlphaFold-predicted structures are used to understand the DVL’s role in Wnt signaling regulation. We also summarize the role of DVL in various diseases and provide insights into further directions for research on the DVL-mediated signaling mechanisms. These findings underscore the importance of DVL as a pharmaceutical target or biological marker in diseases, offering exciting potential for future biomedical applications.

## 1. Introduction

The Wnt signaling pathways play an essential role in numerous developmental processes, hereditary diseases, and various forms of cancer [1,2,3,4,5,6]. The Wnt signaling pathways are broadly categorized into two types: the canonical (β-catenin dependent) pathway and the non-canonical (β-catenin independent) pathway. Canonical Wnt signaling controls cell fate, cell proliferation, embryonic development, stem cell maintenance, tissue homeostasis, and gene expression regulation by regulating the cellular β-catenin level [7,8,9]. Non-canonical Wnt signaling controls cell movement, tissue morphogenesis, planar cell polarity (PCP), and calcium (Ca^2+^) signaling [1,10,11,12,13] (Figure 1).

### 1.1. Canonical Wnt Signaling Pathway

The Wnt canonical signaling pathway exhibits two distinct states: inactivated and activated. Without Wnt ligands, β-catenin is targeted for degradation by the destruction complex, a multi-protein assembly comprising Axin, CK1, APC, and GSK3β. Specifically, CK1 and GSK3β sequentially phosphorylate β-catenin at specific serine/threonine residues. This phosphorylation primes β-catenin for ubiquitination and subsequent proteasomal degradation, maintaining low cytoplasmic β-catenin levels (Figure 1A). This tight regulation prevents aberrant cell proliferation and tumorigenesis. Upon Wnt ligand binding to Frizzled (FZD) receptors and LRP5/6 co-receptors, Dishevelled (DVL) proteins are recruited and activated [14,15,16]. DVL then interacts with Axin, a core component of the destruction complex, and inhibits its activity. This inhibition prevents the further phosphorylation and degradation of β-catenin, leading to its cytoplasmic accumulation. Subsequently, β-catenin moves into the nucleus, where it interacts with TCF/LEF transcription factors to activate the transcription of Wnt target genes. The dysregulation of the Wnt target gene expression can lead to uncontrolled cell proliferation and tumor development.

### 1.2. Non-Canonical Wnt Signaling Pathway

The non-canonical Wnt signaling pathway has been divided into two signaling pathways: Wnt/Ca^2+^ and planar cell polarity (PCP) signaling. Generally, non-canonical Wnt signaling is initiated by the binding of non-canonical Wnt ligands, such as Wnt5a [17] and Wnt 11 [18], to FZD receptors and receptor tyrosine kinase-like orphan receptor 2 (ROR2) [19] or Ryk [20], leading to receptor dimerization and the recruitment of DVL. The recruitment of DVL activates heterotrimeric G-proteins, activating phospholipase C (PLC). The activation of PLC leads to the cleavage of membrane-bound phosphatidylinositol-4,5-bisphosphate (PIP_2_) into inositol-1,4,5-trisphosphate (IP_3_) and diacylglycerol (DAG). IP_3_ induces the release of intracellular calcium ions (Ca^2+^=), resulting in an increased concentration of Ca^2+^ within the cell. This elevated Ca^2+^ concentration activates calcium-sensitive enzymes, including protein kinase C (PKC), calmodulin-dependent protein kinase II (CaMKII), and calcineurin [1,2,3,4,5,6,7,8,9]. The activation of these enzymes subsequently triggers a cascade that activates several transcription factors, such as nuclear factor κB (NF-κB), cAMP response element-binding protein (CREB), and nuclear factor of activated T cells (NFAT), ultimately regulating gene expression and influencing cellular biological functions. The planar cell polarity (PCP) pathway, a branch of non-canonical Wnt signaling, governs coordinated cell polarization within a tissue plane (Figure 1B). This pathway involves other core PCP proteins, such as van Gogh-like (Vangl), Prickle, and Flamingo, which interact with DVL and FZD to establish cell polarity [21]. PCP signaling is crucial for processes like convergent extension movements during embryonic development and the orientation of hair follicles and cilia in epithelial tissues [13].

### 1.3. Dishevelled-Mediated Wnt Signaling

Dishevelled (three DVL paralogs in mammals or single DSH in *Drosophila*) mediates both canonical and non-canonical Wnt signals by transmitting Wnt signals from the membrane-bound Wnt receptors, primarily FZD (Frizzled), to downstream effectors (Figure 1C) [22,23,24,25,26]. Since DVLs are involved in both Wnt signaling pathways, the mechanisms by which DVL regulates the Wnt signaling pathways have been of significant interest [2,3,14,23,27,28,29,30,31,32,33]. However, many questions remain unanswered.

Herein, we focused on human DVL proteins, DVL1/2/3, which have different protein lengths: hDVL1 (695 aa), hDVL2 (736 aa), and hDVL3 (716 aa) (Figure 2A) [25,34]. Among them, hDVL2 accounts for 80–95% of total DVL protein expression. The researchers showed that individual DVL isoforms in mammals may function as a network with some common features and some other more unique ones [25,34,35,36]. DVL proteins can interact with other proteins that regulate the Wnt signaling pathways and communicate with other signaling pathways [3,24,26,37,38,39]. Post-translational modifications, such as the acetylation, phosphorylation, and ubiquitination of DVL functions, have been of great interest [40,41].

This review presents recent advances and insights into the mechanisms of DVL’s action in Wnt signaling pathways, especially at the plasma membrane level. The experimental and AlphaFold-predicted structures [43] of DVL proteins are presented (Figure 2, Figure 3 and Appendix A). AlphaFold2, the second version of AlphaFold, offers highly accurate protein structure prediction [43]. The predicted local distance test (pLDDT) scores, spanning from 0 to 100, are used to determine the confidence of each residue in the predicted structure. Additionally, the predicted aligned error (PAE) estimates the distance error between every pair of residues, providing the confidence level regarding the relative positions and orientations of different parts of the predicted model, such as domains [44,45,46]. The value of pLDDT and PAE maps can be used to understand protein dynamics [44,45,46]. Recently, an AlphaFold3(AF3) study was published, addressing the limitation of AF2 with better accuracy [45,46]. The experimentally determined conserved domains of DVL proteins and the AF3-predicted structure of DVL provide insights into the structure-function relationship of DVL proteins and guide further studies on the mechanism of DVL proteins’ action in the Wnt signaling pathways and aid in the development of new drugs targeting DVL-related diseases [23,47,48,49].

## 2. Structural Insights of Dishevelled

Human DVL proteins contain highly conserved domains such as N-terminal DIX (Dishevelled and Axin), the central PDZ (Post-synaptic density protein-95, Disk large tumor suppressor, Zonula occludens-1) domain, and the C-terminal DEP (Dishevelled, Egl-10, and Pleckstrin) domains (Figure 2) [26]. Each domain’s structure has been characterized using NMR and X-ray crystallography techniques [50,51,52,53]. The structure–function relationship of each domain has been studied. Generally, DVL DIX and PDZ domains activate Wnt/β-catenin, and DVL PDZ and DEP domains activate Wnt/PCP signaling. It has been shown that the DEP domain is also involved in the Wnt/β-catenin pathway [54,55,56]. The functional roles of DVL PDZ and DVL DEP domains in regulating Wnt signaling pathways remain not fully understood [29,30,57]. DVL proteins also contain three intrinsically disordered regions (IDR1, IDR2, and IDR3), which are related to the flexibility of DVL protein (Figure 2) [31,58]. The basic, proline-rich, and extreme C-terminus are also conserved in IDR regions. Recent studies revealed that the roles of IDRs of DVL have been uncovered, such as the autoinhibition of DVL proteins [29,30] and liquid–liquid phase separation (LLPS) [59]. The full-length structure of human DVL proteins has not been resolved experimentally due to the technical challenges in recombinant expression, large intrinsically disordered regions, and post-translational modifications of DVLs [58,60,61]. These challenges may be eventually resolved based on new techniques combining computational modeling such as AlphaFold. Herein, we analyze the predicted human DVL protein structures generated by AF2 or AF3 to explore the structural properties of full-length DVL proteins (Figure 2B and Appendix A) [43,62].

The AF3-predicted hDVL3 structure suggests several interesting structural features (Figure 2B,C): (i) The PAE map of the hDVL3 structure implies that the PDZ and DEP domains may contact each other in the inactive state (Figure 2B,C). Also, the DIX and PDZ domains may be near each other, but this possibility is low due to the high predicted error (low confidence). However, the cloudiness of these domains may be related to the inactive state of DVL. This implies that conformational changes of DVL proteins may be necessary for Wnt signal activation because each domain may interact with its binding partner [29,30,31,32,33]. (ii) Two putative helical structures (IDR2_H1 and IDR3_H1) may be formed within IDR regions. The pLDDT value of the IDR2_H1 is less than 90 but higher than 70, indicating the predicted structure’s high accuracy. However, the pLDDT value of the IDR3_H1 is below 50, leaving the presence of this helix uncertain. (iii) The last C-terminal tail of DVL may bind to the PDZ domain with a very low pLDDT value, implying the low possibility of autoinhibited conformation of DVL (Figure 2B,C). However, Lee et al. [29] demonstrated through biophysical in vitro binding assays that DVL adopts the auto-inhibited conformation through the DVL PDZ and the IRD3_C interaction (called inactivate state). This suggests that the experimental validation of AF predictions is necessary to strengthen the conclusion. In the next section, we discuss the details of the highly conserved domains and IDR regions in DVL proteins.

### 2.1. Conserved Structural Domains of Dishevelled

DVL proteins contain the highly conserved domains DIX, PDZ, and DEP (Figure 3A). Each domain has unique structural properties. The following are the details of the three conserved domains.

The DIX domain is located at the N-terminal of DVL with about 80 amino acid residues (Figure 3B) [63]. The X-ray structure of the DVL DIX domain shows that it consists of a five-stranded β-sheet and an α-helix [51]. A notable feature of the DVL DIX domain is that DVL’s head-to-tail polymerization significantly enhances its local concentration, facilitating efficient signal transduction [15,38,50,51,63]. The binding affinities of homodimer DVL2 DIX and those of heterodimer DVL2 DIX and AXIN1 DIX interaction have been measured experimentally and theoretically [64,65,66,67]. Regarding the DIX domain of DVL interactions, DVL2 DIX-DVL2 DIX and DVL3 DIX-DVL3 DIX homodimers show much weaker interactions than DVL1 DIX-DVL1 DIX. Heterodimer interactions, such as DVL1/2/3 DIX with AXIN1 DIX, are generally stronger than DVL2/3 DIX homodimers but weaker than AXIN1 DIX homodimers, highlighting their distinct roles in modulating Wnt signaling [67]. This heterotypic DVL DIX and AXIN DIX interaction is critical to initiate the Wnt/β-catenin signaling at the plasma membrane [65].

DVL PDZ domain is a conserved protein–protein interaction module, which consists of six β-strands and two α-helices (Figure 3C) [52,68]. DVL PDZ domain features a distinct binding pocket that recognizes the specific C-terminal motifs [69] and small molecules [49,70,71,72], including cholesterol [73,74]. DVL PDZ recognizes the class I PDZ motif, (S/T)-x-Φ (hydrophobic residue), such as Dapper, which is an antagonist of the canonical Wnt signaling, and the class II PDZ motif, Φ-x-Φ, at the C-terminus region of target proteins, such as RYK [75], which activates the non-canonical Wnt signaling. DVL PDZ also recognizes the class III PDZ motif, D/E-x-Φ. Interestingly, the last C-terminal region (IDR3_C, NPSEFFVDVM) of DVL resembles the class III PDZ-binding motif (Figure 2A) [29,30]. The interaction of the DVL1 PDZ domain and the DvlC peptide derived from the last C-terminal region of DVL was verified by using biophysical methods [29]. In addition, the DVL PDZ domain recognizes the internal sequence of peptide motifs [52,76,77]. For example, Omble et al. [78] discovered that DVL2 activates the Wnt/PCP pathway through the interaction between DVL2 PDZ and WGEF (weakly similar to RhoGEF 5/TIM), also known as Arhgef19/Ephexin2. The researchers found that the internal sequence of WGEF (^349^GSTFSLWQDIP^359^) of WGEF, binds to the DVL2 PDZ domain [78]. The results suggested the flexibility of the DVL PDZ domain’s binding pocket to accommodate various ligands [49,78,79]. Interestingly, DVL PDZ-binding small molecules and peptides inhibited the Wnt/β-catenin signaling at the DVL level in cells [49,72,77,80] but activated the Wnt/PCP signaling [29]. The molecular mechanism by which the DVL PDZ domain regulates the canonical and non-canonical Wnt signaling remains incomplete [29,81]. Mieszczanek et al. [81] investigated the role of the PDZ domain of *Drosophila* DSH using CRISPR/Cas9. Their findings suggest that while the PDZ domain is dispensable for canonical Wingless (Wg) signaling, it is crucial in various cellular contexts controlled by non-canonical signaling.

The DVL DEP domain consists of three α-helices, a β-hairpin composed of two β-strands, and two short β-strands, including a finger loop with K435 (amino acid numbering from human DVL3) [53] (Figure 3D). Generally, the primary functions of the DVL DEP domain are as follows: (i) DVL proteins localize to the plasma membrane via the electrostatic interactions between a cluster of basic residues on the DEP domain and acidic lipids, such as PIP_2_ (or PtdIns(4,5)P_2_) in the plasma membrane (Figure 3D) [28,55,82]. This interaction is modulated by the intracellular pH levels, which are partially affected by the Na^+^/H^+^ exchanger NHE2 [28]. (ii) DEP domain can interact with upstream component effectors, such as FZD proteins [22,83,84,85] (Figure 3E and Appendix A). The DVL DEP domain is necessary for WNT-5B-mediated RAC1 activation and the subsequent activation of PCP-JNK signaling [86]. Several groups reported that the DEP domain mainly mediates the FZD/DVL interaction of DVL [22,54,85,87]. Bowin et al. [22] reported that Wnt simulation promoted conformational changes in FZD5-DVL2 and FZD5-DEP complexes using bioluminescence resonance energy transfer (BRET) assays. Hillier et al. [87] reported the cryo-EM structures of FZD3 in complex with extracellular and intracellular binding nanobodies (Nb), showing that Nb9 binds to the cytoplasmic region of FZD3 at the putative DVL or G protein-binding site. Remarkably, Qian et al. [85] reported that the cryo-EM structure of FZD4 engaged with the DVL2 DEP domain, showing that the DEP finger-loop inserts into the FZD4 cavity to form a hydrophobic interface. DEP engages the dimeric receptor as a monomer [85]. Consistent with the experimental results, the AF3 predicts the complex structure of FZD TMD (transmembrane domain) and DVL DEP with high accuracy (Figure 3E and Appendix A, and Appendix A), although the structural differences between experimental and theoretical DEP domain in the complex are recognized (Appendix A). Since the DEP domain binds to the plasma membrane, it will be interesting to see how DEP interacts with FZD and PIP_2_. (iii) DEP-dimerizing to cross-link DIX domain oligomers for signalosome assembly may be important in assembling Wnt signalosomes and mediating signal directionality [54,57]. This domain swapping appears to be regulated by the phosphorylation of specific serine residues (S418 and S435 in DVL2) [57].

### 2.2. Intrinsically Disordered Regions (IDRs) of DVL Proteins

Recent studies have focused on the three intrinsically disordered regions (IDRs) in Dishevelled (DVL) proteins: IDR1, IDR2, and IDR3 (Figure 1, Figure 3A and Appendix A) [59]. IDR1 region is located between the DIX and PDZ domains, IDR2 is between the PDZ and DEP domains, and IDR3 follows the DEP domain (Figure 3A). Since IDRs lack a stable tertiary structure and adopt dynamic conformation under physiological conditions, the IDRs of DVL predicted by AF2 (AF3) showed exceptionally low pLDDT values (Appendix A) [46]

The IRD1 region contains many positively charged residues, including arginine (Arg) and lysine (Lys) residues (IRD1_basic1 and IRD1_basic2) (Figure 2A and Figure 3A). Kang et al. [59] showed that the charged residues in the IDR1 region (spanning amino acids 141–259) of DVL2 are essential for DVL liquid–liquid phase separation (LLPS). LLPS is a cellular process that enables the compartmentalization of proteins and nucleic acids into micron-scale, liquid-like structures within cells. These membraneless organelles, recently termed biomolecular condensates, perform specific cellular functions [88,89]. DVL2 LLPS is essential for forming protein droplets, which in turn facilitates the recruitment of both DVL2 and AXIN1 to the plasma membrane [59]. DVL2 LLPS is crucial for destroying the organization and function of the destruction complex.

AF3 predictions of DVLs indicate that the IDR1 region of full-length DVL may lack a specific structure in its inactive state. However, when analyzing the IDR1 sequence alone, AF3 predicts that the IDR1_basic2 region (^212^RLMRRHKRRRRKQKVSR^228^ in human DVL3) may adopt a helical structure (Figure 3F and Appendix A). Given the basic nature of the IDR1 region, we reasoned potential interactions with the acidic plasma membrane. Using the residue-specific membrane association propensities (ReSMAP) webserver [90], we analyzed the membrane-association propensity (p) of the IDR1_basic2 region. The result indicates that IDR1_basic2 has a higher *p*-value than the IDR1_basic1 region (Figure 3E), suggesting a stronger propensity for binding to the acidic membrane. This membrane interaction could be related to the induction of the IDR1_basic2 region predicted by AF3. Furthermore, since the IDR1_basic2 region is proximal to the DVL PDZ domain, its recruitment to the plasma membrane might influence the interaction between the PDZ domain and the Frizzled (FZD) receptor [52]. These hypotheses warrant further investigation to fully understand the structural dynamics and functional implications of DVL’s intrinsically disordered regions in Wnt signaling and related cellular processes.

The IDR2 region is located between the PDZ and DEP domains (Figure 3A). It has been reported that the IDR2 region of DVL proteins might interact with kinases via the IDR2_proline-rich region [31,58]. AlphaFold predicts that the IDR2-H1 region adopts an α-helical structure with high confidence (90 > pLDDT > 70) (Appendix A). Thus, the structural function of IDR2-H1 in Wnt signaling pathways will be interesting.

IDR3 is the third IDR region of DVL proteins after the DEP domain. The IRD3 region shows evolutionary conservation but displays unique features among DVL isoforms. For example, DVL3 contains the histidine-rich region (Figure 2A). Mutagenesis reveals that the C-terminal sequence of DVL is essential for puncta formation [59]. Recent studies revealed the important roles of the IDR3 regions in the regulation of canonical and non-canonical signaling: (i) The YHEL motif, situated approximately 60 residues downstream of the DEP domain in DVL2 (Figure 2A), interacts with the C-terminal region of the µ2 subunit of the clathrin adaptor protein complex AP-2. This bipartite interaction between DVL and AP-2 is essential for the clathrin-mediated endocytosis of FZD and DVL following Wnt activation, playing a critical role in PCP signaling. (ii) The last C-terminus of DVL can bind to the DVL PDZ domain, adopting the closed conformation. Interestingly, the IDR3-H1 region is predicted to have a putative α-helical structure before the PDZ-binding motif. The role of this putative helical structure of DVLs needs to be explored experimentally. (iii) The IDR3_H1 and IDR3_C regions are indispensable for the DVL complexes and condensates [91].

### 2.3. To Be Closed (Inactive) or Open (Active), That Is the Question

Several biophysical methods have confirmed the interaction between the highly conserved C-terminal tail of DVL (IDR3_C) and its own PDZ domain [29]. These findings suggest that DVL proteins adopt an auto-inhibited conformation (Figure 1) [29,30,31]. The autoinhibited conformation of DVL is through the intramolecular interaction confirmed by the FRET experiment [31]. However, the possibility of intermolecular interactions between IDR3_C and other DVL PDZ domains cannot be excluded, implying the potential for DVL oligomerization.

While naïve DVL proteins typically adopt a closed conformation (inactive state), the transition to an open conformation (active state) is crucial for the activation of Wnt signaling [29,30,31]. In *Xenopus* assays, wild-type DVL was found to be less active than DVL variants like Xdd1 (the PDZ domain was deleted), Xdsh-ΔC (the last 7 amino acid residues are truncated), and Xdsh-GFP (C-terminally tagged GFP) in non-canonical Wnt signaling [29]. These results suggest that the C-terminus of DSH positively regulates non-canonical Wnt/PCP signaling through conformational changes in DVL in vivo. Given that Xdsh-ΔC, Xdd1, and Xdsh-GFP enhance PCP signaling, the researchers hypothesized that co-injecting Xdsh with its binding partners would also enhance PCP signaling (Figure 1C). As expected, the incubation of wild-type Xdsh with a small molecule or TMEM88 peptide enhanced Wnt/PCP signaling in cell-based assays, likely due to conformational changes in DVL [29,30]. While these studies highlight the effects of DVL conformational changes in vivo, the precise mechanisms behind these changes.

It has been reported that phosphorylation of DVL by kinases, such as Casein kinase 1ε (CK1ε), CK2α, and protein kinase C (PKC)δ, is a key sign of activation in both canonical and non-canonical Wnt pathways (Table 1). However, the detailed mechanism of how kinases regulate Wnt signaling remains unclear. These kinases may bind to DVL proteins and induce conformational changes of DVL in vivo, regulating the specific Wnt signaling [58,60,92,93]. Harnoš et al. [31] investigated the conformational dynamics of human DVL3 protein under native conditions using a fluorescein arsenical hairpin binder (FlAsH)-based FRET in vivo approach. Their findings suggest that the IDR regions of DVL3 may be involved in interactions with CK1ε, leading to DVL3 phosphorylation [58]. Biophysical methods further showed that the CK1ε phosphorylation of the DVL PDZ domain disrupts its interaction with the C-terminal end of DVL, resulting in an open conformation of the protein [94]. Recent studies reveal that CK1ε and NIMA-related kinase 2 (NEK2) act as scaffold proteins, regulating DVL conformational dynamics through PDZ domain phosphorylation and modulation of its interaction with the extreme C-terminal tail [58,93]. Interestingly, the last 13 amino acids (IDR3_H1 and IDR3_C) conserved across all DVL isoforms may bind to the third intracellular loop of FZD receptors, stabilizing the FZD-DVL interaction in Wnt/β-catenin signaling.

**Table 1 cells-13-01870-t001:** Kinases and phosphatases interact with DVL.

Kinases	Description	Refs.
CK1 (Casein Kinase I)	CK1 is generally considered to be the key kinase for phosphorylating DVL.	[58,60,92]
CK2 (Casein Kinase II)	CK2 phosphorylates DVL in Drosophila. The phosphorylation site is identified in the same basic PDZ region as CK1.	[95]
PAR-1 (Protease-activated receptor 1)	PAR1 binds and phosphorylates the 36 amino acid segments, DM5, located at the N-terminal end of the PDZ domain of DVL, promoting the Wnt pathway and inhibiting the DVL-mediated JNK pathway.	[96]
Abl (Abelson family Kinases)	Tyrosine kinase Abl binds and phosphorylates DVL on Tyr473 within the DEP domain.	[97]
PLK1 (Polo-like Kinase 1)	DVL2 binds to and phosphorylates the mitotic kinase Polo-like kinase 1 (PLK1) at Thr206, and this phosphorylation is required for spindle orientation and stable microtubule (MT)-KT attachment.	[98]
RIPK4 (Receptor-Interacting Serine/Threonine-Protein Kinase 4)	Receptor-interacting protein kinase 4 (RIPK4), necessary for keratinocyte differentiation, interacts with DVL and is like CK1; it leads to an increase in the Wnt/β-catenin signaling pathway.	[99]
NEK2 (NIMA-Related Kinase 2)	DVL phosphorylated by NEK2 mediates the translocation of connexin from centrosomes during the G2/M phase of the cell cycle.	[93]
PTEN (Phosphatase and tensin homolog)	During ciliogenesis, PTEN controls the serine-143 phosphorylation of DVL2, which serves as a direct substrate of PTEN in vitro.	[100]
PP2A (phosphoprotein phosphatase-2A)	PP2A directly dephosphorylates DVL2 by associating with the DEP domain, which is enhanced by Wnt3a treatment, and this binding suppresses PP2A phosphatase activity and increases Wnt/β-catenin signaling.	[101]
PP1c (Protein phosphatase 1 catalytic subunit)	PP1c binds to and dephosphorylates DVL with the help of homeodomain-interacting protein kinase 2 (Hipk2) to spare DVL from ITCH-mediated ubiquitination and to increase DVL stability and persistence of Wnt signaling.	[102]

## 3. Dishevelled Modification

Post-translational modifications (PTMs) are essential covalent changes that regulate the activity, localization, and stability of proteins, significantly influencing their roles in signaling pathways, such as Wnt signaling [103]. One of the most significant and widespread PTMs is phosphorylation, which is mediated by specific enzymes known as kinases and phosphatases. DVL proteins’ activity and stability are regulated by post-translational modification [40]. In DVL proteins, approximately 15% of the amino acid sequence consists of serine, threonine, and tyrosine residues that can be phosphorylated [60]. So far, over 50 phosphorylation sites have been identified in DVL. Following Wnt binding to its receptor, extensive phosphorylation occurs in the central region of DVL, including the PDZ and DEP domains [58,82]. This phosphorylation is crucial for activating both canonical and non-canonical Wnt signaling pathways. Various kinases, such as Casein Kinase 1 (CK1δ/ε), CK2, PAR-1, Abl, RIPK4, and NEK2, have been shown to phosphorylate DVL under different conditions (Table 1). Additionally, Hanáková et al. [58] identified TTBK2 and Aurora A as novel kinases that target DVL. Besides kinases, several phosphatases [104], such as PP2A, PP5, and PTEN, interact with DVL to dephosphorylate it, thereby regulating its subcellular localization and function (Table 1). Beyond phosphorylation, other PTMs, such as ubiquitination, also play a pivotal role in DVL regulation, particularly in protein degradation and signaling turnover.

Ubiquitination, another important PTM, is mediated by E3 ubiquitin ligases and usually targets proteins for proteasomal or lysosomal degradation [105]. This well-known mechanism regulates various cellular processes, including protein degradation, protein–protein interactions, endocytosis, cell cycle progression, and changes in substrate activity [106]. Deubiquitinating enzymes usually target Lys residues and can reverse the ubiquitination process. DVL proteins play a central role in the Wnt signaling pathway, and their ubiquitination is a critical mechanism for regulating their function (Table 2). The ubiquitination of DVL proteins is located in their N-terminal and C-terminal regions. The ubiquitination of DVL can facilitate its aggregation in the plasma membrane and nucleus [107,108,109]. Vamadevan et al. [109] reported that the IDR1 region of DVL2 is essential for DVL2 phase separation in vitro and in cells, which is essential for forming DVL2 condensates. Deubiquitination, mediated by enzymes like USP, is crucial for maintaining the homeostasis of Wnt signaling and preventing aberrant DVL degradation [110]. Therefore, the dysregulation of DVL deubiquitylation can disrupt the Wnt signaling pathway, leading to aberrant cell proliferation, differentiation, and migration, ultimately contributing to disease development, including cancer. The ubiquitination and deubiquitylation of DVL play a critical role in modulating its function, a process controlled by several factors. Numerous studies suggest that various substances can exert positive or negative regulatory effects on their function by affecting DVL ubiquitination (Table 2) [111,112,113,114].

Furthermore, acetylation is a reversible post-translational modification that alters protein structure and function by adding an acetyl group to the ε-amino group of lysine residues in target proteins [115,116]. Sharma et al. [117] identified acetylation as a novel PTM regulating DVL nuclear localization. The researchers identified 12 acetylation sites on lysine residues of the DVL-1 protein by LC-MS/MS analysis. Acetylation at two key residues, K69 (within the DVL1 DIX domain) and K285 (within the DVL1 PDZ domain), significantly increased the nuclear localization of DVL1 while decreasing its cytoplasmic localization. Subsequent studies showed that K34 within the DIX domain can also be acetylated [108,117]. This finding suggests that acetylation may enhance DVL’s transcriptional regulatory roles, expanding its functional repertoire. Shen et al. [116] reported that the acetylation of K68 in the DVL2 DIX by CREB-binding protein (CBP) modulates DVL2 puncta formation in canonical Wnt signaling activation. They also reported that SIRT2 is responsible for DVL2 deacetylation. Wu et al. [41] investigated and identified DVL3 with more than 50% sequence coverage, including six phosphorylation sites, four methylation sites, and one dimethylation site.

**Table 2 cells-13-01870-t002:** Substances are partially capable of affecting DVL ubiquitination and deubiquitination.

Substances	Description	Refs.
NEDD4L (E3 Ubiquitin Ligase)	NEDD4L induces DVL2 ubiquitination by linking the Lys-6, Lys-27, and Lys-29 amino acids of DVL2.	[118]
CYLD (Deubiquitinating Enzyme)	CYLD recognizes the K63-linked ubiquitin on DVL and binds to DVL, resulting in its deubiquitination.	[119]
Usp14	Usp14 regulates the deubiquitination of DVL by acting on the RDU domain through K63-linked ubiquitin.	[110]
ITCH (A HECT-containing Nedd4-like Ubiquitin E3 Ligase)	ITCH regulates the degradation of DVL by promoting its ubiquitination.	[120]
Prickle-1	Prickle-1 promotes DVL3 degradation.	[121]
Dpr1	Dpr1 promotes pVHL-induced ubiquitination of DVL2.	[122]
RNF185 (E3 Ubiquitin Ligase)	RNF185 enhances ubiquitination and degradation of DVL2.	[123]
HECW1 (E3 Ubiquitin Ligase)	HECW1 promotes DVL1 ubiquitination.	[124]
Huwe1	Huwe1 promotes the ubiquitination of DIX structural domains in DVL.	[125]

## 4. DVL-Related Diseases: Cancers and Other Diseases

DVL proteins are often dysregulated in cancer and are frequently overexpressed in a variety of cancer types [48,86,126,127,128,129,130,131]. The overexpression of DVL promotes cancer cell invasion [26,132] and contributes to tumor progression by driving uncontrolled cell growth, increasing metastatic potential, and influencing the tumor microenvironment. Given their critical role in these oncogenic processes, DVL proteins have been identified as a biological marker and a potential target for cancer therapy [48].

### 4.1. Roles in Cancer

**Non-small cell lung cancer (NSCLC)**: Lung cancer, which originates from the lung tissue, encompasses various types, such as non-small cell lung cancer (NSCLC) and small cell lung cancer (SCLC). These tumors have the potential to metastasize to other organs, including the brain, being particularly susceptible to secondary tumors. NSCLC accounts for about 85% of all malignant pulmonary tumors, which are frequently identified in the later stages when invasion and metastasis occur [133]. A comprehensive study conducted by Kafka et al. [134] investigated the expression levels of key proteins involved in the Wnt signaling pathway, specifically DVL1, DVL3, E-cadherin (CDH1), and β-catenin (CTNNB1), in brain metastases originating from primary lung carcinomas. Their findings revealed the significant overexpression of DVL1 and DVL3 in 87.1% and 90.3% of brain metastasis samples, respectively. The researchers concluded that the alterations in the expression of DVL1, DVL3, E-cadherin, and β-catenin in brain metastases underscore the importance of the Wnt signaling pathway in the metastatic process of lung cancer to the brain, providing valuable insights into potential therapeutic targets for intervention.

Wei et al. [135] confirmed the important role of PWP1(periodic tryptophan protein 1) in NSCLC. Mechanistic studies demonstrated that PWP1 activates the Wnt signaling pathway through interaction with DVL2 and inhibits the Hippo signaling pathway by suppressing Merlin, thereby promoting the proliferation and invasion of NSCLC cells. Zhao et al. [136] showed that DVL1 is overexpressed in NSCLC tissues, and the nuclear co-expression of DVL1 with β-catenin proteins is significantly associated with poor patient prognosis. The results suggest that DVL1 is a potential therapeutic target for NSCLC and that miR-214 inhibits the DVL1-mediated Wnt/β-catenin signaling pathway in NSCLC cells. DVL3 also affects the malignant phenotype of NSCLC through the non-canonical Wnt-PCP pathway but not the canonical Wnt pathway in NSCLC cells (A549, SPC, H157, H460, and LTE) [36]. In a Wnt5B ligand-dependent manner, the recruitment of DVL3 via the DEP domain to the membrane for phosphorylation activates JNK signaling [86]. Lee et al. [39] identified TMEM88 as a DVL-binding protein. Zhang et al. [137] also demonstrated that membrane-bound TMEM88 inhibits the Wnt signaling pathway, while cytoplasmic TMEM88 promotes epithelial-mesenchymal transition (EMT) and enhances the invasiveness of lung cancer cells. Wang et al. [137] showed that the interaction between DVL1 and CtBP2 modulates the Wnt/β-catenin signaling pathway, influencing cell proliferation and drug sensitivity in NSCLC. The results indicate that CtBP2 overexpression is linked to adverse clinical outcomes in NSCLC, suggesting its potential as a prognostic biomarker.

Cui et al. [138] reported that DVL3 is upregulated in patients with PDGN-LUAD (Pan-driver-gene-negative lung adenocarcinoma). Ji et al. [139] focused on the antitumor mechanism of SPATA2 in non-small cell lung cancer (NSCLC), particularly its regulatory effect on the DVL1 protein. Their research found that SPATA2 inhibits the ubiquitination of DVL1, thereby disrupting its role in the Wnt/β-catenin signaling pathway, which inhibits epithelial-mesenchymal transition (EMT). Zhao et al. [86] focused on the mechanism of Wnt5B in NSCLC. Through in vitro and in vivo experiments, the research team confirmed that Wnt5B promotes the invasion and metastasis of NSCLC cells by activating the Wnt/PCP signaling pathway via Frizzled3 (FZD3) and DVL3. These studies suggest that multiple proteins in the DVL family play important roles in NSCLC. DVL1 is associated with poor prognosis, DVL2 promotes tumor proliferation and invasion through interaction with PWP1, and DVL3 drives cancer cell invasion and metastasis via Wnt5B-regulated signaling. Together, these findings highlight the multifunctional roles of the DVL protein family in NSCLC progression, suggesting their potential as therapeutic targets.

**Colorectal Cancer (CRC)**: Aberrant Wnt signaling activation is a defining feature of colorectal cancer (CRC), with DVL playing a key role in promoting tumor progression. Zheng et al. [140] explored the role of the long-stranded non-coding RNA testis-specific transcript Y-linked 15 (TTTY15) in CRC and found that its expression was significantly upregulated in CRC tissues. Its expression was positively correlated with DVL3 and negatively correlated with the expression of miR-29a-3p. Similarly, Li et al. [141] investigated the role of circular RNA_0101802 (circ_0101802) in CRC. They found that it was also significantly upregulated in CRC tissues with a positive correlation to DVL3 expression and negative correction to miR-665 expression. Shen et al. [116] reported that acetylated K68 DVL2 (DVL2-aK68) levels were significantly increased during the transition from normal to stages I-II. In CRC, the deubiquitination enzyme USP14 has been reported to deubiquinate DVL2 at the K444 and K451 sites, reinforcing Wnt signaling [110]

Yin et al. [142] studied FUBP1(Far Upstream Element Binding Protein 1), a multifunctional protein primarily known for regulating gene expression in CRC. They found that FUBP1 activated the Wnt/β-catenin pathway by binding to the *DVL1* promoter, which enhanced the expression of stem cell markers like *c-Myc*, *NANOG*, and *SOX2*. Knocking down DVL1 significantly suppressed the stemness and tumorigenicity of CRC cells, suggesting that FUBP1 plays a critical role in CRC progression via DVL1. Tang et al. [143] focused on the role of DVL2 in colitis-associated colorectal cancer (CAC) and found that DVL2 interacted with TNF receptor 1 (TNFRI), facilitating its endocytosis and inhibiting NF-κB signaling. The pro-inflammatory cytokine IL-13 upregulated DVL2 expression via STAT6, activating Wnt signaling and promoting CAC development. Targeting STAT6 effectively reduced DVL2 levels and reduced cancer cell proliferation, emphasizing the key role of DVL2 in CAC progression.

**Intracranial meningiomas**: Intracranial meningiomas are tumors that develop from the meninges, the protective membranes surrounding the brain and spinal cord. Bukovac et al. [144] investigated the role of DVL1 in intracranial meningiomas that arise from the meninges, the protective membranes surrounding the brain and spinal cord. Their study focused on the DVL1 PDZ domain, revealing that certain base repeat deletions differentially affect the function of the DVL1 protein. The nuclear expression of DVL1 was found to correlate significantly with high expression of active β-catenin and meningioma grade, suggesting that DVL1 may serve as a biomarker for meningioma progression and Wnt pathway activation.

**Hepatocellular carcinoma (HCC)**: Hepatocellular carcinoma (HCC), the most common type of primary liver cancer, is often linked to dysregulation in several key signaling pathways, including the Wnt/β-catenin pathway. The aberrant activation of the Wnt/β-catenin signaling occurs in human HCC samples [145] with DVL2 and DVL3 being upregulated in the disease [128]. Given their association with poor prognosis, targeting DVL2 and DVL3 in HCC could offer new therapeutic strategies to inhibit tumor progression and improve patient outcomes. [128,132].

The elevated expression of DVL proteins, particularly DVL2 and DVL3, suggests their major involvement in the development of the tumor [128]. Increased levels of DVL1 and DVL3 proteins have been correlated with poor prognosis, as they are involved in promoting cell survival and proliferation through their impact on the Wnt/β-catenin pathway [128,132]. Zhu et al. [146] identified TM4SF1 (Transmembrane 4 L Six Family Member 1) as a binding protein for DVL2 in HCC. TM4SF1, typically localized at the cell membrane, enhances Wnt/β-catenin signaling by promoting the interaction of DVL2 with Axin. Zhang et al. [147] showed that ASPM promotes HCC progression by inhibiting autophagy-mediated degradation of DVL2, which activates Wnt/β-catenin signaling. Autophagy, a process where cells degrade malfunctioning components, is disrupted by ASPM, which binds to DVL2 and prevents its breakdown, thereby promoting Wnt/β-catenin signaling [147]. The stabilization of DVL2 through ASPM presents a potential target in HCC [147].

Wang et al. [148] and Lo et al. [149] reported that Cripto-1 is overexpressed in roughly 50% of hepatocellular carcinoma (HCC) specimens. Cripto-1 interacts with the FZD7/LRP6 receptor complex and DVL3, resulting in increased stability of DVL3. Such an interaction triggers signaling pathways that promote proliferation, invasion, and metastasis in HCC.

**Esophageal squamous cell carcinoma (ESCC)**: ESCC is a cancer affecting the esophagus, showing that the Wnt pathway is activated [150]. Chen et al. [129] explored the expression of DVL3 in ESCC and its mechanism of action. The results showed that DVL3 promoted the proliferation and migration of ESCC cells, whereas silencing DVL3 significantly inhibited these tumor properties and suppressed tumor formation by promoting apoptosis. Therefore, DVL3 can potentially be a target for the early diagnosis, prognosis, and treatment of ESCC [129]. Fu et al. [151] investigated the role of circular RNA circ_0000277 in ESCC and its potential mechanism. The results showed that circ_0000277 regulates DVL3 expression as a ‘sponge’ for miR-1294, promoting proliferation, migration, and invasion of ESCC cells. In contrast, inhibition of circ_0000277 expression attenuated these malignant properties and showed significant tumor suppression in vivo experiments. Targeting circ_0000277 could decrease DVL3 levels, therefore resulting in the suppression of the aggressive behaviors of ESCC cells and offering a novel approach to treating this cancer. DVL3 appears to be a good target for therapeutic intervention, as evidenced by its direct role in cell proliferation and migration and its control by circ_0000277 and miR-1294 [151]. The DVL3 regulatory axis may be disrupted in future ESCC treatments to reduce the spread of cancer and enhance patient outcomes as research continues to provide insight into the complexity of this signaling system. 

**Cervical Cancer**: HECW1 (HECT, C2, and WW domain-containing E3 ubiquitin protein ligase 1) is an E3 ubiquitin ligase that plays a role in protein ubiquitination and degradation [152,153]. By tagging proteins for proteasome breakdown, ubiquitination is a biological mechanism that regulates the quantity of specific proteins in a cell. The role of HECW1, an E3 ubiquitin ligase, is to aid in the attachment of ubiquitin molecules to target proteins, designating them for degradation [152,153]. Xu et al. [124] reported that HECW1 inhibits cervical cancer cell proliferation and tumor formation by promoting the ubiquitination of DVL1 and down-regulating the Wnt/β-catenin signaling pathway. HECW1 affects the protein expression of DVL1, a potent activator of Wnt/β-catenin. Inhibition of HECW1 suppresses the ubiquitination of DVL1, thereby upregulating its expression.

**Breast Cancer**: The abnormal activation of the Wnt signaling has been observed in many breast cancers, contributing to uncontrolled cell proliferation and tumor growth [154]. Rasha et al. [155] found that DVL2 controls cancer cell proliferation and T cell-mediated immunity in HER2-positive breast cancer through the DVL2 loss of function studies, implying the potential immune regulatory role of DVL2 proteins. Alongside tumor growth, this shows that DVL affects the immune system in the tumor microenvironment [155,156]. Due to its ability to enhance immune responses and slow tumor progression, targeting DVL2 offers a dual therapeutic advantage [155]. Jiang et al. [157] reported that Wnt5b is a key regulatory factor that controls the basal-like breast cancer phenotype. This cancer type is characterized by aggregate behavior, stem-like phenotype, and poor clinical outcomes influenced by activating canonical and noncanonical Wnt signaling.

**Gastric Cancer**. He et al. [130] reported that DVL2 promotes gastric cancer progression through the Wnt/β-catenin pathway. Using cell lines and 209 gastric cancer specimens, they investigated the role of DVL2 in gastric cancer. DVL2 overexpression plays a crucial role in the occurrence and development of gastric cancer. DVL2 expression was significantly correlated with many clinicopathological parameters. The overexpression of DVL2 promotes gastric cancer initiation and progression mainly through aberrant activation of the Wnt/β-catenin signaling pathway. The depletion of endogenous DVL2 inhibited the proliferation, migration, and invasion of gastric cancer cells. Therefore, DVL2 may become a new therapeutic target for gastric cancer. The overall survival of patients with high expression of DVL2 was significantly shorter than those with low expression. Therefore, DVL2 may become a new therapeutic target for gastric cancer [130].

Across multiple cancer types, DVL proteins are frequently overexpressed, contributing to uncontrolled proliferation, invasion, and metastasis. These findings highlight DVL as a critical target for therapeutic intervention.

### 4.2. Roles in Other Diseases

DVL’s involvement in Robinow syndrome (RS) [158,159,160,161,162,163,164] and aging and neurodegenerative diseases [111,165,166,167,168], such as Parkinsonism and Alzheimer’s disease, highlights its broader physiological importance beyond cancer. The following are the details.

**Robinow Syndrome (RS)**: Robinow syndrome (RS) is a complex disorder characterized by widespread abnormalities affecting the skeletal and urogenital systems, as well as neurological impairments [158]. Genetic studies have revealed that mutations in the DVL1 and DVL3 genes can lead to the autosomal dominant form of RS [159,160,161,162]. Gignac et al. [163] conducted insightful mechanistic studies using the *Drosophila* and chicken model to elucidate the functions of DVL1 in dominant RS. They expressed the variant forms of DVL1 alongside the endogenous genome of *Drosophila* and chicken, showing that DVL1 variants caused a weakened canonical and an enhanced non-canonical Wnt signaling in several assays, which is consistent with the *Xenopus* assays by other researchers [29,30]. The results indicate that the IDR3 region of DVLs alters the balance between different Wnt signaling pathways. Tophkheane et al. [164] also investigated DVL1 variants and C-terminal deletions, reporting that DVL variants have differential effects on craniofacial development and Wnt signaling. This result showed that the C-terminus of DVL is important to both branches of Wnt signaling [29,30].

**Parkinsonism.** Dysregulation of DVL-mediated Wnt signaling has been implicated in neuronal loss [167]. Parkinsonism is characterized by the degeneration of dopaminergic neurons in the substantia nigra [168]. Shen et al. [111] focused on the critical role of DVL-2 in Parkinson’s disease (PD). The research found that inhibiting von Hippel–Lindau (VHL) protein-mediated ubiquitination of DVL-2 enhances the stability of DVL-2 and exerts neuroprotective effects via the DVL-2/β-catenin axis. The inhibition of VHL led to reduced degradation of DVL-2, promoting β-catenin activity and alleviating the degeneration of dopaminergic neurons. The study also designed a competitive peptide, Tat-DDF-2, to inhibit the interaction between VHL and DVL-2, showing potential pharmacological effects in preventing Parkinson’s disease. These findings indicate that DVL2 plays a significant regulatory role in the pathogenesis and treatment of PD [111].

## 5. Conclusions

DVL proteins precisely control Wnt signaling pathways, interacting with many effectors. In canonical Wnt signaling, DVL forms large molecular complexes involving Wnt, FZD, and LRP5/6 [14,16,59,65,66,167]. This promotes the disassembly of the β-catenin destruction complex, β-catenin accumulation, and Wnt signaling activation [169]. Recent structural studies have provided insights into the Wnt-FZD-LRD5/6 complex using cryo-EM spectroscopy [16]. Future studies on the structural and functional properties of the XWNT8-FZD8-LRP6-DVL-AXIN complex and the potential for DVL-targeted therapeutics could yield transformative insights into Wnt signaling mechanisms and disease treatment strategies [170]. In non-canonical Wnt signaling, DVL protein interacts with FZD protein and other proteins, such as ROR [10,19] and RYK proteins [11,20]. The structural basis of the DVL-mediated non-canonical Wnt pathway, especially Wnt/Ca^2+^ signaling, has yet to be explored intensively, although there was evidence from the bioassays [171]. An intriguing area for further study is understanding how DVL transitions from its auto-inhibited (inactive) conformation to an active state. The molecular mechanisms underlying the activation process remain unclear. The roles of DVL-binding partners and LLPS need to be explored more extensively. Since CRISPR/Cas9 can edit the gene of DVL paralogs, the roles of endogenous DVL proteins in Wnt signaling in normal and cancerous cells in vivo will be revealed. Given DVL’s involvement in various diseases [48], developing targeting therapeutics holds promise [47]. While most efforts have focused on the DVL PDZ domain [49,69,70,71,72,172,173] (Appendix A), exploring compounds targeting other conserved domains, intrinsically disordered regions, or DVL-associated kinases could serve as valuable probes or tools for modulating specific Wnt signaling pathways. Despite recent advances in DVL-mediated Wnt signaling, even more exciting discoveries are likely on the horizon.

## Figures and Tables

**Figure 1 cells-13-01870-f001:**
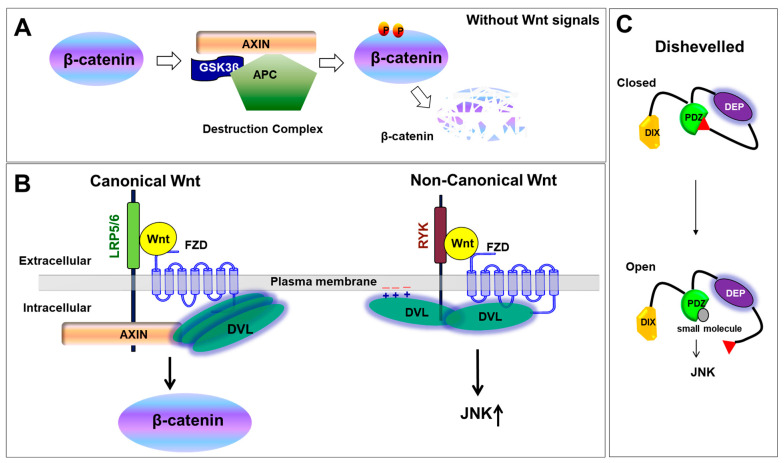
Plausible mechanism of DVL-mediated Wnt signaling pathways. (**A**) The destruction complex degrades the β-catenin protein. DVL is inactivated through autoinhibition. (**B**) DVL regulates both canonical and non-canonical Wnt signaling through its interaction with Frizzled (FZD) and other co-receptors. Upon ligand binding, DVL inhibits the β-catenin degradation complex in canonical signaling or activates downstream effectors in non-canonical signaling. (**C**) Conformational changes of DVL may regulate Wnt signaling.

**Figure 2 cells-13-01870-f002:**
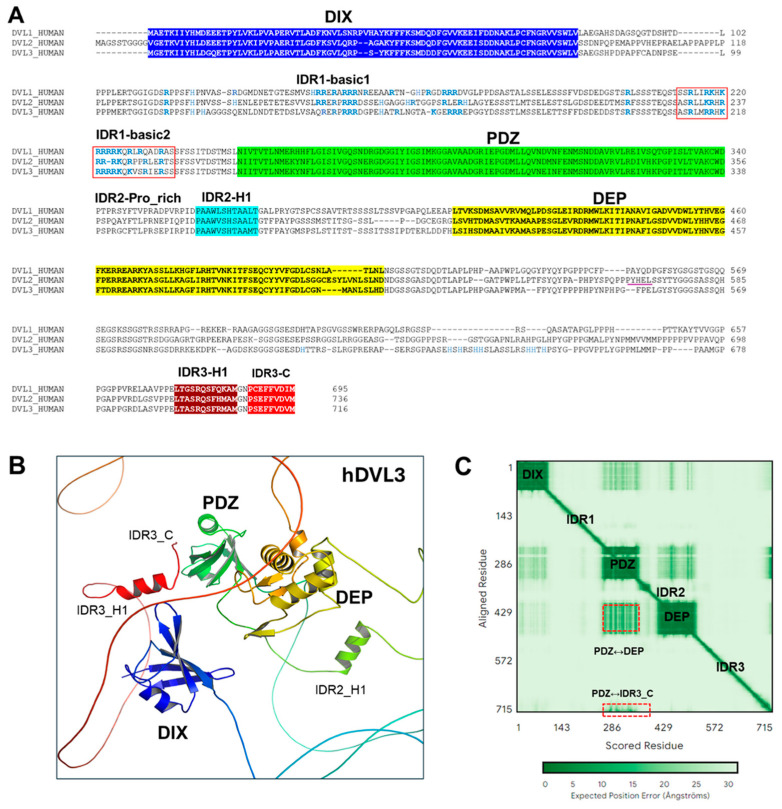
Human DVL proteins have three paralogs, DVL1/2/3. (**A**) Sequence alignment of human DVL proteins using ClustalW [42]. The conserved DIX (blue), PDZ (green), and DEP (yellow) domains are highlighted. The intrinsically disordered regions (IDRs) are indicated. The red box represents the IDR1-basic 2 region. (**B**) AlphaFold-predicted DVL3 structure shows the autoinhibition conformation. In the intrinsically disordered regions, AF3 predicts two putative helical structures. IDR2_H1 is located between the PDZ and DEP domains. IDR3_H1 is near the C-terminal region of DVL. (**C**) Predicted aligned error (PAE) maps for AF3-predicted human DVL3 protein structure.

**Figure 3 cells-13-01870-f003:**
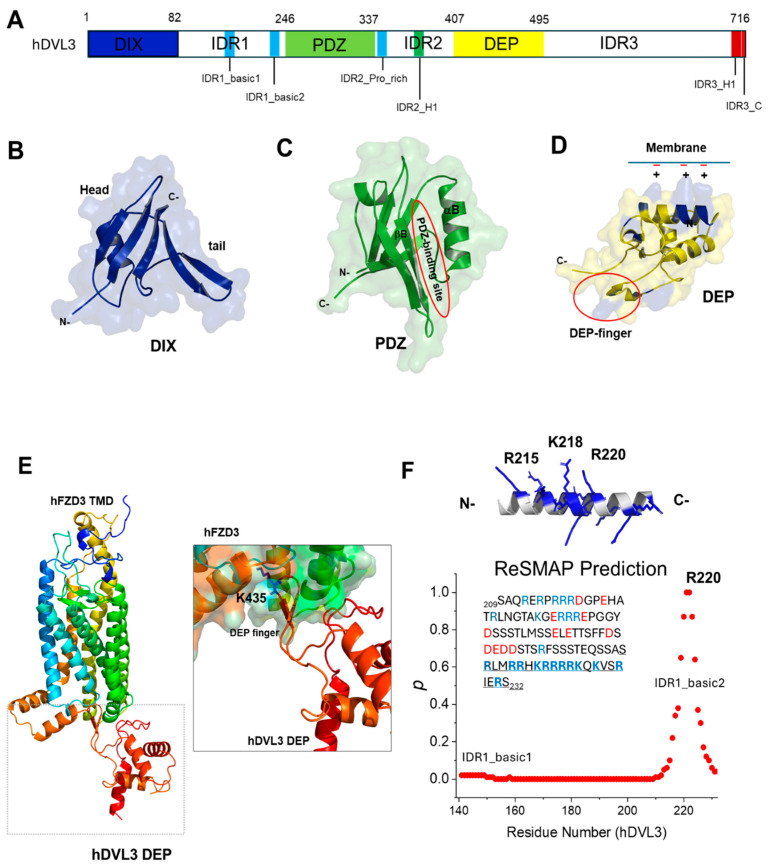
The structures of the highly conserved domains and IDR1 region of hDVL3. (**A**) Structure diagram of hDVL3 protein (UniProt ID: Q92997). (**B**–**D**) AlphaFold-predicted structure of the highly conserved domains: DIX, PDZ, and DEP domain. (**B**) The DVL DIX domain can interact with other DIX domains, such as Axin, through head and tail interfaces. (**C**) PDZ domain is a protein-protein interaction module. The red circle represents the PDZ-binding site. More than 30 binding partners for the PDZ domain were reported [3,24,37]. (**D**) The DEP domain can interact with the plasma membrane through its positively charged surface with the negatively charged lipids, such as PIP_2_ of the inner layer of the plasma membrane. The red circle represents the DEP-finger. (**E**) AF3-predicted the complex structure of the hFZD3 transmembrane domain (TMD) with the hDVL3 DEP domain. The binding site of hFZD3 TMD and hDVL3 DEP was expanded. The residues in the binding interfaces, hydrogen bonds, and salt bridges of the FZD3-DVL3 DEP complex predicted by AF3 are summarized in Appendix A and Appendix A. The DEP-finger region may bind to the intracellular region of hFZD3. (**F**) AF3-predicted structure of IDR1_basic2 region of hDVL3, showing the putative α-helical structure. ReSMAP prediction shows that IDR1_basic2 of DVLs may bind to the plasma membrane (Appendix A shows the IDR1_basic region of hDVL1).

## Data Availability

Not applicable.

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
