# Peer review of "Structural and Functional Insights into Dishevelled-Mediated Wnt Signaling"

_cells, 2024, doi:10.3390/cells13221870_

Round 1

Reviewer 1 Report

Comments and Suggestions for Authors

The manuscript Structural and Functional Insights into Dishevelled-Mediated Wnt Signaling  provide a clear and uptodate synthesis of the involvement of DVL in Wnt signaling. 

I have minor comments. Some are indicated in the manuscript.

Line 81: DVL has 3 paralogs in human. The authors should give some information about the presence of paralogs in other species. What is known?

Lines 543-573: Wnt is also implicated in Aging and Alzeihmer disease. What is known about the involvement of DVL in these pathologies?

Palomer, E., Buechler, J., Salinas, P.C., 2019. Wnt Signaling Deregulation in the Aging and Alzheimer's Brain. Frontiers in Cellular Neuroscience 13, 227.

Kostes, W.W., Brafman, D.A., 2023. The Multifaceted Role of WNT Signaling in Alzheimer's Disease Onset and Age-Related Progression. Cells 12.

Author Response

Reviewer 1

The manuscript Structural and Functional Insights into Dishevelled-Mediated Wnt Signaling provide a clear and up-to-date synthesis of the involvement of DVL in Wnt signaling. 

[Comment 1] I have minor comments. Some are indicated in the manuscript.

[Response 1] We appreciate the reviewer's comments on the manuscript. We modified them as the reviewer suggested.  

[Comment  2] Line 81: DVL has 3 paralogs in human. The authors should give some information about the presence of paralogs in other species. What is known?

[Response 2] We added more information, such as the following:  

Dishevelled (three DVL paralogs in mammals or single DSH in Drosophila) mediates both canonical and non-canonical Wnt signals by transmitting Wnt signals from the membrane-bound Wnt receptors, primarily FZD (Frizzled), to downstream effectors (Figure 1C) [22-26].

[Comment  3] Lines 543-573: Wnt is also implicated in Aging and Alzeihmer disease. What is known about the involvement of DVL in these pathologies?

Palomer, E., Buechler, J., Salinas, P.C., 2019. Wnt Signaling Deregulation in the Aging and Alzheimer's Brain. Frontiers in Cellular Neuroscience 13, 227.

Kostes, W.W., Brafman, D.A., 2023. The Multifaceted Role of WNT Signaling in Alzheimer's Disease Onset and Age-Related Progression. Cells 12.

[Response 3] We appreciate the reviewer’s suggestion. Since we focused on cancer, we briefly added the Dishevelled’s roles in other diseases.

4.2. Roles in other diseases 

DVL's involvement in Robinow syndrome (RS) [158-164] and aging and neurodegenerative diseases [111, 165-168], such as Parkinsonism and Alzheimer’s disease, highlights its broader physiological importance beyond cancer. The following are the details.  

Reviewer 2 Report

Comments and Suggestions for Authors

1. I suggest that you simplify the description of the interactions between the DIX, PDZ, and DEP domains.

2. I suggest that you emphasize the need for experimental validation of AlphaFold predictions to strengthen the conclusions.

3. I suggest that you expand the discussion to include other signaling pathways like Wnt/Ca2+ to provide a more balanced view of DVL's roles.

4. I suggest that you add visual aids showing DVL’s transition between inactive and active conformations to clarify the autoinhibition mechanism.

5. I suggest that you include more specific future research directions, especially regarding potential therapeutic targets related to DVL’s intrinsically disordered regions.

6. I suggest that you expand on how ubiquitination and other post-translational modifications influence DVL’s role in diseases, particularly cancer.

Author Response

[Comment 1] I suggest that you simplify the description of the interactions between the DIX, PDZ, and DEP domains.

[Response 1] We discussed the interactions between the DIX, PDZ, and DEP domains, as a reviewer suggested. Lines 135-142

The AF3-predicted hDVL3 structure suggests several interesting structural features (Figure 2B and 2C): (i) The PAE map of the hDVL3 structure implies that the PDZ and DEP domains may contact each other in the inactive state (Figure 2B and 2C). Also, the DIX and PDZ domains may be near each other, but this possibility is low due to the high predicted error (low confidence). However, the cloudiness of these domains may be related to the inactive state of DVL. This implies that conformational changes of DVL proteins may be necessary for Wnt signal activation because each domain may interact with its binding partner [29-33].

[Comment 2] I suggest that you emphasize the need for experimental validation of AlphaFold predictions to strengthen the conclusions.

[Response 2] We agreed. We mentioned a reviewer’s comment in the modified manuscript.

Lines 151-152: This indicates that the experimental validation of AF predictions is necessary to strengthen the conclusion.

[Comment 3] I suggest that you expand the discussion to include other signaling pathways like Wnt/Ca2+ to provide a more balanced view of DVL's roles.

[Response 3] While the role of DVLs in Wnt/Ca2+ has been reported, only a few studies are available. Further studies of Dvl’s role in Wnt/Ca2+ signaling are needed. So, we mentioned the further study of DVL’s role in this signaling in conclusion. (Lines 631).

[Comment 4] I suggest that you add visual aids showing DVL’s transition between inactive and active conformations to clarify the autoinhibition mechanism.

[Response 4] We added the model of the autoinhibition mechanism of DVL in Figure 1C.

[Comment 5] I suggest that you include more specific future research directions, especially regarding potential therapeutic targets related to DVL’s intrinsically disordered regions.

[Response 5] In conclusion, we mentioned the future research directions related to DVL’s IDRs regions. Most studies of DVL have focused on the DVL PDZ domain (SI Table S2).  

[Comment 6] I suggest that you expand on how ubiquitination and other post-translational modifications influence DVL’s role in diseases, particularly cancer.

[Response 6] We already discussed the roles of deubiquitination and ubiquitination of DVL in lines 381-386. Also, we added some information to the manuscript as follows: Line 474: In CRC, the deubiquitination enzyme USP14 has been reported to deubiquinate DVL at K444 and K451 sites, reinforcing Wnt signaling [110]

Reviewer 3 Report

Comments and Suggestions for Authors

The manuscript by Wang and colleagues summarized the role of Dishevelled (DVL) in Wnt signaling pathways and various human diseases. The novelty of this review lies in the prediction of DVL3 structure and the complex structure of DVL3 with its interacting partner hFZD3 via AlphaFold. By using the AlphaFold predicated models, the authors potentially shed light on the regulatory roles of DVLs in the Wnt signaling pathways. However, there are some concerns that need to be further addressed. 

Major

Line 14-15, in the abstract, the authors stated that the AlphaFold-predicated DVL structures were used to understand the mechanisms of DVLs in Wnt signaling pathways. The predicted models are intrinsically not experimental structures and can not be used as solid structures to reveal their mechanisms. Even though the highly conserved DIX, PDZ and DEP domains were predicted with high confidence, the disordered regions with low predicated score are likely to present significant discrepancies.  Therefore, the predicated models can only be used to facilitate the understanding of DVL’s role in Wnt signaling regulation.

Line 82, Figure 1a, which should be Figure 2A.

Line 109, X-ray spectroscopy techniques, which should be X-ray crystallography based on the references.

Line 131, DVL3 protein, which should be DVL3 structure.

In Figure 3B, the label DIX overlayed with the structure. The authors should label it carefully.

In Figure 3E, the rotation of the model looks weird. The authors should check it carefully.

Line 144, cellular region, which should be intracellular region.

In Figure 3F, the authors should label the residues with name and number.

Line 150-151, the authors stated that “This implies that conformational changes of DVL proteins may be involved as Wnt activation”, which needs additional evidence.

Based on the predicated model of the DVL3-FZD3 complex, what types of interaction are involved between the DLV3 and FZD3 interface? The authors should analyze it in more details.

Minor

Line 24-26, text redundancy with line26-30, which can be revised as “…into two types: the canonical (β-catenin dependent) pathway and the non-canonical (β-catenin dependent) pathway. “

Line 119, The complete structure of human DVL proteins. In structural biology, the full-length structure of human DVL proteins is commonly used.

Line 608, incomplete sentence.

Numerous grammar errors need to be carefully corrected. For example:

Line 78, both Wnt signaling, which should be both Wnt signalings.

Line 113, regulating Wnt signaling pathways, which should be in regulating Wnt signaling pathways.

Author Response

The manuscript by Wang and colleagues summarized the role of Dishevelled (DVL) in Wnt signaling pathways and various human diseases. The novelty of this review lies in the prediction of DVL3 structure and the complex structure of DVL3 with its interacting partner hFZD3 via AlphaFold. By using the AlphaFold predicated models, the authors potentially shed light on the regulatory roles of DVLs in the Wnt signaling pathways. However, there are some concerns that need to be further addressed. 

[Response 0] Thank you for the comments from a reviewer. The comments are very helpful in improving our manuscript to be published. The following are our responses to each comment from a reviewer.

[Comment 1] Line 14-15, in the abstract, the authors stated that the AlphaFold-predicated DVL structures were used to understand the mechanisms of DVLs in Wnt signaling pathways. The predicted models are intrinsically not experimental structures and can not be used as solid structures to reveal their mechanisms. Even though the highly conserved DIX, PDZ and DEP domains were predicted with high confidence, the disordered regions with low predicated score are likely to present significant discrepancies.  Therefore, the predicated models can only be used to facilitate the understanding of DVL’s role in Wnt signaling regulation.

[Response 1] We appreciate the reviewer’s comments. We modified the abstract as suggested.

The experimentally determined conserved domain structures of DVL in conjunction with AlphaFold-predicted structures are used to understand the DVL’s role in Wnt signaling regulation.

[Comment 2] Line 82, Figure 1a, which should be Figure 2A.

[Response 2] We fixed it.

[Comment 3] Line 109, X-ray spectroscopy techniques, which should be X-ray crystallography based on the references.

[Response 3] We fixed it.

[Comment 4] Line 131, DVL3 protein, which should be DVL3 structure.

[Response 4] We fixed it.

[Comment 5] In Figure 3B, the label DIX overlayed with the structure. The authors should label it carefully.

[Response 5] We fixed it.

[Comment 6] In Figure 3E, the rotation of the model looks weird. The authors should check it carefully.

[Response 6] We modified Figure 3E. We wanted to show the binding of DEP to the intracellular region of Frizzled. We put detailed information on the complex structure of FZD3 and hDVL3 DEP in SI Table S1 and SI Figure S2.

[Comment 7] Line 144, cellular region, which should be intracellular region.

[Response 7] Appreciated. We fixed it.

[Comment 8] In Figure 3F, the authors should label the residues with name and number.

[Response 8] We labeled the residues with their names and numbers in Figure 3E as suggested.

[Comment 9] Line 150-151, the authors stated that “This implies that conformational changes of DVL proteins may be involved as Wnt activation”, which needs additional evidence.

[Response 9] The conformational changes of DVL have been examined using the fluorescence method. We put the references. We add some comments to the manuscript.

The AF3-predicted hDVL3 structure suggests several interesting structural features (Figure 2B and 2C): (i) The PAE map of the hDVL3 structure implies that the PDZ and DEP domains may contact each other in the inactive state (Figure 2B and 2C). Also, the DIX and PDZ domains may be near each other, but this possibility is low due to the high predicted error (low confidence). The cloudiness of these domains may be related to the inactive state of DVL. This implies that conformational changes of DVL proteins may be necessary for Wnt signal activation because each domain may interact with its binding partner [29-33].

[Comment 10] Based on the predicated model of the DVL3-FZD3 complex, what types of interaction are involved between the DLV3 and FZD3 interface? The authors should analyze it in more details.

[Response 10] We analyzed the complex structure of DVL3 and FZD3 interface with PDBePISA website (SI Table S1 and Figure S2C).

[Comment 11] Line 24-26, text redundancy with line26-30, which can be revised as “…into two types: the canonical (β-catenin dependent) pathway and the non-canonical (β-catenin dependent) pathway. “

[Response 11] We revised the sentences as a reviewer’s suggestion. The following is the modified version.

The Wnt signaling pathways are broadly categorized into two types: the canonical (β-catenin dependent) pathway and the non-canonical (β-catenin independent) pathway. Canonical Wnt signaling controls cell fate, cell proliferation, embryonic development, stem cell maintenance, tissue homeostasis, and gene expression regulation by regulating the cellular b-catenin level [7-9].

[Comment 12] Line 119, The complete structure of human DVL proteins. In structural biology, the full-length structure of human DVL proteins is commonly used.

[Response 12] Appreciated. We fixed it.

[Comment 13] Line 608, incomplete sentence.

[Response 13] We fixed it. The authors declare no conflicts of interest.

[Comment 14] Numerous grammar errors need to be carefully corrected. For example: Line 78, both Wnt signaling, which should be both Wnt signalings. Line 113, regulating Wnt signaling pathways, which should be in regulating Wnt signaling pathways.

[Response 14] Appreciated. We fixed them.

Round 2

Reviewer 2 Report

Comments and Suggestions for Authors

No more comments